# Pre-Stroke Statin Therapy Improves In-Hospital Prognosis Following Acute Ischemic Stroke Associated with Well-Controlled Nonvalvular Atrial Fibrillation

**DOI:** 10.3390/jcm10143036

**Published:** 2021-07-08

**Authors:** Paweł Wańkowicz, Jacek Staszewski, Aleksander Dębiec, Marta Nowakowska-Kotas, Aleksandra Szylińska, Agnieszka Turoń-Skrzypińska, Iwona Rotter

**Affiliations:** 1Department of Medical Rehabilitation and Clinical Physiotherapy, Pomeranian Medical University in Szczecin, Żołnierska 48, 71-210 Szczecin, Poland; aleksandra.szylinska@gmail.com (A.S.); agi.skrzypinska@gmail.com (A.T.-S.); iwrot@wp.pl (I.R.); 2Department of Neurology, Military Medical Institute, Szaserów 128, 04-141 Warszawa, Poland; jacekstaszewski@wp.pl (J.S.); adebiec@wim.mil.pl (A.D.); 3Department of Neurology, Medical University of Wrocław, Borowska 213, 50-566 Wrocław, Poland; marnow64@interia.pl

**Keywords:** ischemic stroke, atrial fibrillation, NOAC, VKA, statin, outcome, mortality

## Abstract

Many studies have confirmed the positive effect of statins in the secondary prevention of ischemic stroke. Although several studies have concluded that statins may also be beneficial in patients with atrial fibrillation-related stroke, the results of those studies are inconclusive. Therefore, the aim of this study was to analyze the effect of pre-stroke statin therapy on atrial fibrillation-related stroke among patients with a well-controlled atrial fibrillation. This retrospective multicenter analysis comprised 2309 patients with acute stroke, with a total of 533 patients meeting the inclusion criteria. The results showed a significantly lower neurological deficit on the National Institutes of Health Stroke Scale at hospital admission and discharge in the group of atrial fibrillation-related stroke patients who took statins before hospitalization compared with those who did not (*p* < 0.001). In addition, in-hospital mortality was significantly higher in the atrial fibrillation-related stroke patients not taking statins before hospitalization than in those who did (*p* < 0.001). Based on the results of our previous research and this current study, we postulate that the addition of a statin to the oral anticoagulants may be helpful in the primary prevention of atrial fibrillation-related stroke.

## 1. Introduction

Atrial fibrillation (AF) is the main cardiac rhythm disorder. An inadequate treatment or late detection may result in serious health complications such as ischemic stroke (IS). Currently, the prevalence of AF in the general population is age-dependent and varies from 4% to 15% [1,2,3]. As a result of systematic developments in medicine and the consequent increase in the lifespan of the general population, these percentages are expected to increase dramatically in the next decade [4,5].

Old-generation oral anticoagulants (vitamin K antagonists, VKAs) or non-vitamin K antagonist oral anticoagulant (NOACs) are the most popular therapeutic options used to prevent embolic events in patients with AF. The use of VKAs reduces the risk of IS by 65% and reduces subsequent mortality by 25% compared with a placebo group [6,7,8]. The main reason behind failures in this therapy are reductions in the international normalized ratio (INR), which can affect its efficacy and increase the risk of bleeding complications. These difficulties were the reason for initiating research on NOACs. These drugs have at least the same efficacy in the prevention of IS as VKAs, but at a reduced risk of life-threatening hemorrhages compared with VKAs, relaxing the requirement for continuous monitoring of blood levels [9,10,11,12,13,14,15].

Non-modifiable AF risk factors include gender, age, ethnicity, and genetic factors, while modifiable risk factors include heart failure, coronary artery disease, hypertension, atherosclerosis, or diabetes [16,17,18,19,20]. Significantly, all these AF risk factors are also recognized risk factors for IS [21]. Despite this, the latest ESC guidelines for the diagnosis and management of AF, developed in collaboration with the European Association for Cardio-Thoracic Surgery (EACTS), still mainly focus on the embolic mechanism and the implementation of an oral anticoagulant in the presence of an appropriate constellation of modifiable and non-modifiable risk factors based on the CHADS2, CHA2DS2-VASc, or ABC pathway, but with marginal attention paid to prothrombotic, atherogenic, and proinflammatory risk factors [22]. In the literature, only a few studies have addressed the co-occurrence of embolic and prothrombotic risk factors in IS patients with coexisting AF.

In an earlier study, we examined the configuration of ischemic stroke risk factors in patients with nonvalvular atrial fibrillation (NVAF) and therapeutic INR levels. Based on a univariable and multivariable logistic regression model, we found that there were more smokers (OR = 20.337; OR = 147.589), patients with a previous ischemic stroke (OR = 6.556; OR = 11.094), patients with hypertension (OR = 3.75; OR = 2.75), and patients with dyslipidemia (OR = 2.318; OR = 2.294) with these factors [23]. In a following multicenter study, we examined the configuration of IS risk factors in patients with NVAF that had been treated with NOACs, finding that AF patients treated with the NOACs displayed higher thrombotic, proatherogenic, and proinflammatory risk factors, in addition to the embolic risk closely associated with AF [24]. Based on the aforementioned observations, statins appear to be ideal candidates to complement the anticoagulant effects of VKAs and NOACs, with a broad-spectrum of positive clinical effects on thrombotic, proatherogenic, and proinflammatory factors. Numerous studies in diverse patient populations have shown that statins (HMG-CoA reductase inhibitors) exhibit effects beyond just lowering lipid levels—they improve endothelial dysfunction, increase nitric oxide bioavailability, have antioxidant properties, inhibit the inflammatory response, and stabilize atherosclerotic plaque. Moreover, an increasing number of studies also mention the positive role of statins in the recruitment of endothelial progenitor cells, immunosuppressive activity, or inhibition of myocardial hypertrophy [25,26,27]. A large meta-analysis has confirmed the positive effect of statins in the secondary prevention of IS [28]. Although several studies have concluded that statins may also be beneficial in patients with AF-related stroke, the results of those studies are inconclusive [29,30,31,32]. Therefore, the aim of this study was to analyze the effect of pre-stroke statin therapy on AF-related stroke among patients with a well-controlled AF.

## 2. Materials and Methods

This retrospective multicenter analysis comprised 2309 patients hospitalized at three Neurology Clinics in Poland (Department of Neurology at the Pomeranian Medical University in Szczecin, Department of Neurology at the Military Medical Institute in Warsaw, and Department of Neurology at the Medical University in Wroclaw) with acute ischemic stroke between 2014 and 2020.

Inclusion criteria for the study were as follows: (1) aged 18 years or older, (2) NVAF treated with VKA and a therapeutic INR (2.0–3.0) on admission, (3) NVAF treated with a NOACs, and (4) information regarding the use of statins before the acute IS. Exclusion criteria were as follows: (1) no data regarding the use, or not, of statins before acute IS; (2) no or irregular intake of NOACs; and (3) non-therapeutic levels of INR (<2.0).

In total, 533 patients with acute IS, with NVAF treated with NOACs and VKAs, and a therapeutic level of INR were eligible for this study after meeting the inclusion criteria. These patients were divided into two groups based on receiving information on using statins, or not, before their IS. Group one comprised 191 patients who were not using statins, with group two comprising 342 patients who were using statins (Figure 1).

All patients with IS had undergone neurological examination, ECG, neuroimaging, and vascular and biochemical testing according to the guidelines of the American Heart Association and American Stroke Association.

The neurological status of the patients was assessed on the day of admission and on the day of discharge, according to the National Institutes of Health Stroke Scale (NIHSS) used to assess the severity of neurological deficit in a patient with ischemic stroke. The NIHSS was designed around the traditional neurological examination to assesses consciousness, eye movements, visual fields, motor and sensory impairments, ataxia, speech, cognition, and inattention. A score of 0 usually indicates normal functioning, with a higher score indicating the degree of deficit. The individual scores for each item are added together to calculate a total personal NIHSS score. The maximum possible score is 42 and the minimum score is 0. Importantly, a score of 0 does not indicate a valid neurological examination because it does not account for subtle neurological deficits [33].

Basic demographic data such as age, gender, information on common risk factors for IS such as hypertension, dyslipidemia, diabetes, coronary artery disease, peripheral artery disease, hemodynamically significant stenosis of the internal carotid artery, and smoking, as well as information on mortality during hospitalization, were collected on the basis of the medical interview and analysis of medical records. The Bioethics Committee of the Pomeranian Medical University issued a consent to conduct the study (KB-0012/49/07/2020/Z).

Diagnosis of atrial fibrillation was based on either ECG or medical records at admission [34]. The diagnosis of hypertension was based on an analysis of medical records, patient history, and hypotensive medication intake. Hypertension was defined as a systolic blood pressure of 140 mmHg or higher and/or a diastolic blood pressure of 90 mmHg or higher in repeated measurements [35]. The diagnosis of diabetes mellitus was based on an analysis of medical records, patient history, and p/diabetic medication intake. Diabetes was defined as a fasting blood glucose level of 126 mg/dL or higher, or a blood glucose level of 200 mg/dl or higher measured throughout the day [36]. The diagnosis of dyslipidemia was based on an analysis of medical records, patient history, and hypolipemic medication intake. Dyslipidemia was defined as a serum cholesterol concentration of >190 mg/dL, low-density lipoprotein cholesterol > 115 mg/dL, serum triglyceride concentration > 150 mg/dL, and high-density lipoprotein cholesterol < 40 mg/dL in males and <45 mg/dL in females. The diagnosis of coronary artery disease was based on an analysis of medical records, patient history, and cardiovascular medication intake. The diagnosis of significant internal carotid artery stenosis was based on vascular examination (Doppler ultrasound/angio-tomography or angio-resonance imaging) [37,38]. The diagnosis of peripheral artery disease was based on an analysis of medical records, patient history, and intake of cardiovascular medications. The definition of cigarette smoking was determined by current smoking of any number of cigarettes, an analysis of medical records, or from medical interviews with the patient.

### Statistical Analysis

Statistica v13.0 software (StatSoft, Inc., Tulsa, OK, USA) was used to perform the statistical analysis.

The null hypothesis stated that there were no significant improvement in patients with AF-related stroke who had taken statins in the prehospital period. The alternative hypothesis stated that the use of statins in the prehospital period in patients with AF-related stroke does positively affect in-hospital outcomes and mortality. The distribution of the data was tested using a Shapiro–Wilk test. Quantitative data were analyzed using a Mann–Whitney U test. Qualitative data were analyzed based on the *X*^2^ *t*est. If the subgroup size was insufficient, a Yates’s correction was used. The relationship between the analyzed parameters was evaluated using univariable and multivariable logistic regression model analysis. The multivariable logistic regression was corrected for potentially distorting data (age, dyslipidemia, diagnosed hypertension, and smoking). Differences were statistically significant at *p* ≤ 0.05.

## 3. Results

The median age of all patients was 79.0 years (77.9 ± 8.8 years). One hundred and seventy-four patients (32.65%) were male and 359 patients (67.35%) were female. Hypertension was present in 504 patients (94.56%), diabetes mellitus in 239 (44.84%), dyslipidemia in 393 (73.73%), internal carotid artery significant stenosis/occlusion in 64 (12%), coronary heart disease in 338 (63.41%), and peripheral arterial disease in 77 (14.44%), and 242 patients (45.40%) were smokers.

### 3.1. Comparison of Variables in Patients with AF-Related Stroke Who Were Taking Statins before Hospitalization versus Those Not Taking Statins before Hospitalization

Compared with the patients with AF-related stroke who did not take statins before hospitalization, the patients with AF-related stroke who took statins before hospitalization were significantly younger (83 vs. 77 years; *p* < 0.001) and were also much more likely to have hypertension (99.42% vs. 85.86%; *p* < 0.001) and dyslipidemia (100% vs. 26.98%; *p* < 0.001), and to smoke cigarettes (51.75% vs. 34.03%; *p* < 0.001). We also observed a significantly lower neurological deficit on the NIHSS scale at admission and at hospital discharge in the group of AF-related stroke patients who took statins before hospitalization compared with those who did not. We also found that in-hospital neurological improvement was significantly more common in patients with AF-related stroke who took statins before hospitalization (Figure 2). In-hospital mortality was significantly higher in the AF-related stroke patients not taking statins before hospitalization than in those AF-related stroke patients taking statins before hospitalization (50.26% vs. 2.92%; *p* < 0.001). A comparison of both groups is shown in Table 1.

### 3.2. Prognosis in Patients with Acute AF-Related Stroke Who Were Taking Statins before Hospitalization

After adjusting the results for age, hypertension, dyslipidemia, and smoking, we confirmed that the prehospital use of statins in patients with AF-related stroke was related to a reduced risk of death and a milder stroke course. The results are presented in Table 2.

## 4. Discussion

In a previous study, we showed that (a) both non-modifiable and modifiable AF risk factors are also recognized risk factors for IS; and (b) AF patients treated with VKAs and who had therapeutic INR levels, as well as AF patients treated with NOACs, have a higher prevalence of thrombotic, proatherogenic, and pro-inflammatory risk factors. In another study conducted by Yang et al., it was observed that the presence of cardioembolic risk factors was independently associated with left atrial volume index, persistent atrial fibrillation, heart failure, and body mass index. In contrast, the presence of non-cardioembolic risk factors was independently associated with coronary artery calcium score, hypertension, diabetes, and age [39]. This indicates that statins may be the ideal candidates to complement the anticoagulation effects of VKAs and NOACs, with their broad-spectrum positive clinical effect on thrombotic, proatherogenic, and proinflammatory risk factors. However, cardiologists do not recommend the routine use of statins in all patients with atrial fibrillation as a primary prophylaxis for ischemic stroke, rather relying solely on the prescription of older or new-generation oral anticoagulants. Neurologists recommend statins only for the prevention of atherosclerotic strokes [22,40].

The available literature includes studies reporting the beneficial effects of statins on the prognosis of patients with AF-related stroke, as well as studies showing no such association. In a meta-analysis by Biffi et al., the use of statins prior to a stroke is shown to be correlated with better outcomes for small vessel strokes than any other stroke subtype [41]. Another meta-analysis by Eun et al. shows that the use of statins before AF-related stroke is associated with a lower risk of poor short-term functional outcomes [42].

The results of the current study indicate that, among the entire studied group of patients with well-controlled AF, those who were medicating with statins before the onset of IS had a higher prevalence of thrombotic, proatherogenic, and proinflammatory risk factors than those patients not taking statins. In other words, the patients treated with statins were at a higher risk of adverse vascular complications compared with the patients who did not receive such treatment before the onset of IS. Despite the higher prevalence of thrombotic, proatherogenic, and proinflammatory risk factors, those patients treated with statins achieved better NIHSS scores (both at admission and discharge) compared with those patients not taking statins before their stroke. This indicates the pleiotropic effect of statins that goes beyond a mere reduction of lipid levels [43].

Statin therapy in the pre-stroke period increases cerebral collateral circulation, which is crucial in the acute phase of AF-related stroke when a large artery is suddenly blocked by an embolus and the effective collateral circulation is needed rapidly. In this way, statins may reduce the risk of large cortical infarcts and severe neurological deficits usually associated with an AF-related stroke [44]. This is all the more significant as the recanalization in this type of stroke is lower compared with an atherosclerotic stroke [45]. The neovascularization potential of rosuvastatin and atorvastatin has been confirmed in animal models [46].

Previous research investigating the association of pre-stroke statin medication in patients with AF-related stroke has shown better survival in these patients [29,30]. These findings are consistent with the results of our study, in which we demonstrated that, in the entire group of patients with well-controlled AF, those who were not treated with statins before IS had a significantly higher in-hospital mortality compared with those patients medicating with statins in the pre-stroke period (50.26% vs. 2.92%; *p* < 0.001).

The mechanisms by which statins reduce mortality in AF-related stroke are related to their pleiotropic effects on the cardiovascular and endocardial circulatory systems [47,48]. Statins have also been found to improve left ventricular function, reduce levels of interleukin-6 and C-reactive protein, increase nitric oxide bioavailability, increase antioxidant properties, and inhibit inflammatory response and plaque stabilization. Potential mechanisms that may mediate the beneficial effects of statins on the circulatory system include modulation of endothelial function, anti-inflammatory effects, antioxidant properties, stabilize atherosclerotic plaque, and angiogenesis [49]. The cardioprotective potential of statins has been confirmed in a wide range of studies on cardiovascular pathology, in which statins have been shown to inhibit myocardial remodeling, prevent AF and life-threatening ventricular arrhythmias, preserve nitric oxide production in heart failure, reduce small G-protein activity in myocardial hypertrophy, and contribute to myocardial repair after ischemia by mobilizing endothelial progenitor cells and protecting the myocardium from damage [50,51,52].

Our study has several limitations. First, it is a retrospective study. Second, reference to specific types of statins was not possible because of the limited number of cases. Third, the absence of stroke type (embolic or atherosclerotic) data. Fourth, the absence of echo data, such as LA size, LV function, and spontaneous echo contrast (SEC). Fifth, the absence of data on AF temporal type (paroxysmal/persistent). Despite these limitations, our study seems to be the first multicenter study focusing on the effect of pre-stroke medication with statins in a selected group of patients with AF-related stroke who were also taking NOACs or maintaining therapeutic levels of INR.

## 5. Conclusions

Statin therapy before IS is associated with a reduced risk of mortality and improved functional outcome in the acute phase of AF-related stroke. Based on the results of our previous research and this current study, we postulate that the addition of a statin to the oral anticoagulants may be helpful in the primary prevention of AF-related stroke, although further well-designed randomized controlled trials are necessary to confirm this association.

## Figures and Tables

**Figure 1 jcm-10-03036-f001:**
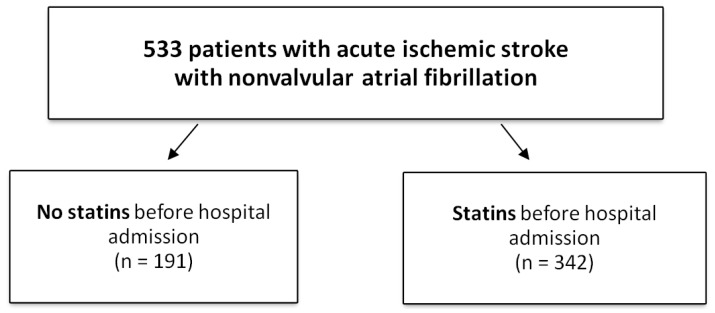
Study flowchart.

**Figure 2 jcm-10-03036-f002:**
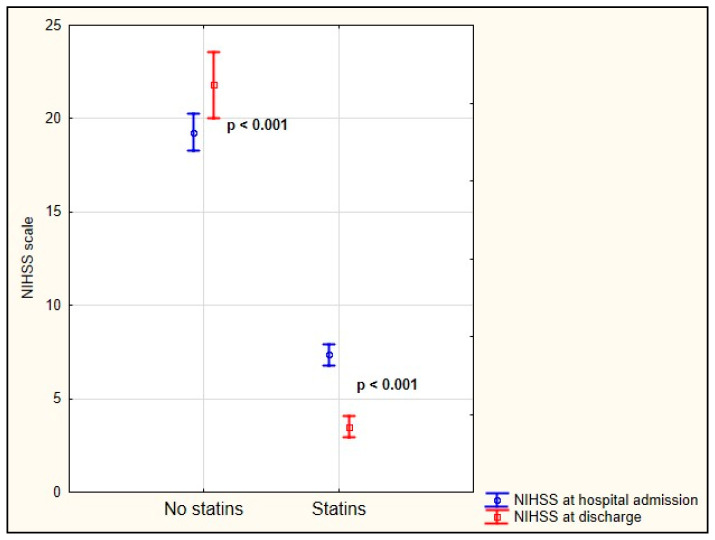
Change in NIHSS scores before hospital admission and at hospital discharge.

**Table 1 jcm-10-03036-t001:** Comparison of patients taking and not taking statins.

Variables.		No Statins before Hospital Admission *n* = 191	Statins before Hospital Admission *n* = 342	*p*
Age [years], mean ± SD; median		81.86 ± 8.67; 83.0	75.76 ± 8.18; 77.0	<0.001
Gender	male	55 (28.80%)	119 (34.80%)	0.157
	female	136 (71.20%)	223 (65.20%)	
Hypertension	no	27 (14.14%)	2 (0.58%)	<0.001
	yes	164 (85.86%)	340 (99.42%)	
Coronary heart disease	no	70 (36.65%)	125 (36.55%)	0.982
	yes	121 (63.35%)	217 (63.45%)	
Peripheral arterial disease	no	165 (86.39%)	291 (85.09%)	0.682
	yes	26 (13.61%)	51 (14.91%)	
Diabetes mellitus	no	106 (55.50%)	188 (54.97%)	0.907
	yes	85 (44.50%)	154 (45.03%)	
Smoking	no	126 (65.97%)	165 (48.25%)	<0.001
	yes	65 (34.03%)	177 (51.75%)	
Dyslipidemia	no	138 (73.02%)	0 (0.00%)	<0.001
	yes	51 (26.98%)	342 (100.00%)	
Stenosis/occlusion	no	167 (87.43%)	302 (88.30%)	0.767
	yes	24 (12.57%)	40 (11.70%)	
NIHSS at hospital admission, mean ± SD; median		19.27 ± 6.71; 20.0	7.36 ± 5.29; 6.0	<0.001
NIHSS at discharge, mean ± SD; median		21.81 ± 12.01; 23.0	3.51 ± 5.29; 2.0	<0.001
Delta NIHSS, mean ± SD; median		3.25 ± 8.52; 4.0	−3.73 ± 3.77; −3.0	<0.001
Death	no	95 (49.74%)	332 (97.08%)	<0.001
	yes	96 (50.26%)	10 (2.92%)	

Abbreviations: *p*—level of statistical significance, *n*—number of patients, Me—median, SD—standard deviation, NIHSS—National Institutes of Health Stroke Scale.

**Table 2 jcm-10-03036-t002:** Multivariate regression model for patients taking statins.

Variables	No Adjusted				Adjusted *			
	*p*	OR	CI OR—95%	CI + OR 95%	*p*	OR	CI OR—95%	CI + OR 95%
Death	<0.001	0.030	0.015	0.059	<0.001	0.022	0.008	0.060
NIHSS on admission to hospital	<0.001	0.766	0.734	0.799	<0.001	0.790	0.742	0.843
NIHSS at discharge	<0.001	0.818	0.791	0.846	<0.001	0.840	0.804	0.877
Delta NIHSS	<0.001	0.824	0.791	0.858	<0.001	0.775	0.718	0.836

Abbreviations: *p*—level of statistical significance, OR—odds ratio, Cl—confidence interval, NIHSS—National Institutes of Health Stroke Scale. Notes: * adjusted by age, hypertension, dyslipidemia, smoking.

## Data Availability

All data that support the findings of this study are available upon request from the corresponding author.

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
