# Peer review of "Pre-Stroke Statin Therapy Improves In-Hospital Prognosis Following Acute Ischemic Stroke Associated with Well-Controlled Nonvalvular Atrial Fibrillation"

_jcm, 2021, doi:10.3390/jcm10143036_

Round 1

Reviewer 1 Report

This is a retrospective observational study with the attendant disadvantages of potential confounding. In the introduction, the authors mention that results of previous studies have been inconclusive.  Sadly, due to its design as an inevitably observational study, this is too.  

The most glaring potential confounder is the difference between the median ages of those treated before their stroke with statins (75.7y) and those not so treated (81.86y).  Since age is perhaps the most powerful determinant of risk for stroke and outcome afterwards, this disparity makes any conclusion or potential therapeutic advice highly suspect.  

It is of interest that the prevalence of hypertension (as defined by the authors) and smoking is greater in the statin-treated group although this may have contributed to higher risk scores leading to more aggressive risk management prior to the study.    

The statin-treated group had all (by definition) been diagnosed as having dyslipidaemia although no other specific criteria are given - apart from a diagnosis in the records. Pre-treatment lipid levels and thresholds for diagnosis and treatment are not given.  

The study is interesting and the hypothesis that pre-treatment of AF patients at risk of stroke with statins to reduce the risk any resulting disability is plausible.  Unfortunately, the confounding factors in this study do not allow any firm conclusion about the benefit of such therapy to be drawn.  Better matching of characteristics of the groups might allow firmer conclusions. 

The paper needs to draw more attention to these weaknesses and avoid firm conclusions for therapy. 

Author Response

Reviewer 1:

Thank you very much for your review and valuable tips. 

  • This is a retrospective observational study with the attendant disadvantages of potential confounding. In the introduction, the authors mention that results of previous studies have been inconclusive.  Sadly, due to its design as an inevitably observational study, this is too.  

Exactly as the reviewer noted, a retrospective study is not an ideal study to link cause and effect. As noted by the reviewer, our study showed confounding data such as age, hypertension, dyslipidemia, and smoking. Nevertheless, in the multivariate regression model - Table 2 and Line 184-187, we excluded these confounders, confirming that the pre-hospital use of statins in patients with AF-related stroke was related with a reduced risk of death and a milder stroke course. In our opinion, this study may be of great help to both the neurological and cardiological communities in determining the appropriate prevention of ischemic stroke in the near future.

  • The most glaring potential confounder is the difference between the median ages of those treated before their stroke with statins (75.7y) and those not so treated (81.86y).  Since age is perhaps the most powerful determinant of risk for stroke and outcome afterwards, this disparity makes any conclusion or potential therapeutic advice highly suspect.  

As we mentioned in the previous answer in multivariate regression model - Table 2 and Line 184-187, we excluded confounding factors including the age highlighted by the reviewer, while confirming that pre-hospital statin use in patients with AF-related stroke was associated with a reduced risk of death and a milder stroke course

  • It is of interest that the prevalence of hypertension (as defined by the authors) and smoking is greater in the statin-treated group although this may have contributed to higher risk scores leading to more aggressive risk management prior to the study.    

The results of this current study indicate that among the entire studied group of patients with well-controlled AF, those who were medicating with statins before the onset of IS had a higher prevalence of thrombotic, proatherogenic and proinflammatory risk factors than those patients not taking statins. In other words, the patients treated with statins were at a higher risk of adverse vascular complications compared to the patients who did not receive such treatment before the onset of IS. Despite the higher prevalence of thrombotic, proatherogenic and proinflammatory risk factors, those patients treated with statins achieved better NIHSS scores (both at admission and discharge) compared to those patients not taking statins before their stroke. This indicates the pleiotropic effect of statins that goes beyond a mere reduction of lipid levels

  • The statin-treated group had all (by definition) been diagnosed as having dyslipidaemia although no other specific criteria are given - apart from a diagnosis in the records. Pre-treatment lipid levels and thresholds for diagnosis and treatment are not given.  

As suggested by the reviewer, we have modified the manuscript Lines 128-131.

  • The study is interesting and the hypothesis that pre-treatment of AF patients at risk of stroke with statins to reduce the risk any resulting disability is plausible.  Unfortunately, the confounding factors in this study do not allow any firm conclusion about the benefit of such therapy to be drawn.  Better matching of characteristics of the groups might allow firmer conclusions. The paper needs to draw more attention to these weaknesses and avoid firm conclusions for therapy. 

As noted by the reviewer, our study showed confounding data such as age, hypertension, dyslipidemia, and smoking. Nevertheless, in the multivariate regression model - Table 2 and Line 184-187, we excluded these confounders, confirming that the pre-hospital use of statins in patients with AF-related stroke was related with a reduced risk of death and a milder stroke course.

Reviewer 2 Report

Major comments

1. Do you have any information on the patient's stroke type (embolic or atherosclerotic)? It is necessary to analyze the effect of statin use according to the stroke type.

Not all strokes in AF patients are embolic. In particular, the patients in this study (Hypertension in 94.56%, DM in 44.84%, dyslipidemia in 73.73%, carotid artery stenosis/occlusion in 12%, coronary heart disease in 63.41%, peripheral arterial disease in 77 14.44%) have a very high risk of atherosclerosis. In addition, because the authors included only AF patients who were using anticoagulants very well in the study, it is very likely that most of the strokes in this study are atherosclerotic strokes rather than embolic strokes. Since most of the strokes were atherosclerotic strokes, wouldn't the use of statins have a positive effect on the outcome?

If an analysis according to the stroke type is not possible, it is a major limitation of this study. Please describe and discuss this in as detail as possible.

2. It was well reported in the following paper that very many atherosclerotic non-cardioembolic risk factors were found despite atrial fibrillation-associated ischemic stroke. “Non-cardioembolic risk factors in atrial fibrillation-associated ischemic stroke. PLoS One.2018; 13(7): e0201062.” It is hoped that the authors discuss the risk of atherosclerotic non-cardioembolic stroke in AF patients and discuss the benefits of statin use in these AF patients.

3. In this study, the absence of echo data, such as LA size, LV function, and spontaneous echo contrast (SEC), and the absence of data on AF temporal type (paroxysmal/persistent) are a major limitation. Previous studies have reported that degree of LA structural remodeling and the presence of SEC are associated with the poor prognosis of stroke with AF. (PMID: 26001853, J Neurol Sci. 2015 Jul 15;354(1-2):97-102. PMID: 27188406, Stroke. 2016;47:1920-1922.) Please mention and discuss these limitations.

Minor comments

1. Line 37-38, “Old-generation oral anticoagulants (vitamin K antagonists, VKAs) or new-generation oral anticoagulants (NOACs) are the most popular therapeutic options”

It is better to label NOAC as "Non-vitamin K antagonist oral anticoagulant" rather than "new-generation oral anticoagulant". Since the abbreviation NOAC is a word commonly used in guidelines and various studies, the unity of terminology is important in research papers.

2. Line 48, “modifiable risk factors include cardiovascular insufficiency,”

It is better to use the term "Heart failure", more commonly used, instead of the term "Cardiac insufficiency".

3. Line 55, “ABC scores”

ABC in ESC guidelines is not a score. The term referring to integrated AF management should be called the ABC pathway.

Author Response

Dear Reviewer, thank you very much for your review and valuable tips. In line with the reviewer's recommendations, we have modified the manuscript.

  1. Do you have any information on the patient's stroke type (embolic or atherosclerotic)? It is necessary to analyze the effect of statin use according to the stroke type. Not all strokes in AF patients are embolic. In particular, the patients in this study (Hypertension in 94.56%, DM in 44.84%, dyslipidemia in 73.73%, carotid artery stenosis/occlusion in 12%, coronary heart disease in 63.41%, peripheral arterial disease in 77 14.44%) have a very high risk of atherosclerosis. In addition, because the authors included only AF patients who were using anticoagulants very well in the study, it is very likely that most of the strokes in this study are atherosclerotic strokes rather than embolic strokes. Since most of the strokes were atherosclerotic strokes, wouldn't the use of statins have a positive effect on the outcome?If an analysis according to the stroke type is not possible, it is a major limitation of this study. Please describe and discuss this in as detail as possible.

Unfortunately, in order to obtain the largest possible research group, we did not specify the type of stroke. Nevertheless, we are currently working on a randomized trial in which we will consider each element, including the type of stroke. As suggested by the reviewer, we have modified the limitation section. Line 251-252. 

  1. It was well reported in the following paper that very many atherosclerotic non-cardioembolic risk factors were found despite atrial fibrillation-associated ischemic stroke. “Non-cardioembolic risk factors in atrial fibrillation-associated ischemic stroke. PLoS One.2018; 13(7): e0201062.” It is hoped that the authors discuss the risk of atherosclerotic non-cardioembolic stroke in AF patients and discuss the benefits of statin use in these AF patients.

As suggested by the reviewer, we have modified the discussion without disturbing the entire narrative of this work. Lines 194-199. We discuss the benefits of statins in patients with atrial fibrillation in lines 206-249.

  1. In this study, the absence of echo data, such as LA size, LV function, and spontaneous echo contrast (SEC), and the absence of data on AF temporal type (paroxysmal/persistent) are a major limitation. Previous studies have reported that degree of LA structural remodeling and the presence of SEC are associated with the poor prognosis of stroke with AF. (PMID: 26001853, J Neurol Sci. 2015 Jul 15;354(1-2):97-102. PMID: 27188406, Stroke. 2016;47:1920-1922.) Please mention and discuss these limitations.

As suggested by the reviewer, we have completely modified the limitation section. Line 250-254.

Minor comments

  1. Line 37-38, “Old-generation oral anticoagulants (vitamin K antagonists, VKAs) or new-generation oral anticoagulants (NOACs) are the most popular therapeutic options”

It is better to label NOAC as "Non-vitamin K antagonist oral anticoagulant" rather than "new-generation oral anticoagulant". Since the abbreviation NOAC is a word commonly used in guidelines and various studies, the unity of terminology is important in research papers.

We have modified the text as suggested by the reviewer. Line 37-38.

  1. Line 48, “modifiable risk factors include cardiovascular insufficiency,”

It is better to use the term "Heart failure", more commonly used, instead of the term "Cardiac insufficiency".

We have modified the text as suggested by the reviewer. Line 48.

  1. Line 55, “ABC scores”

ABC in ESC guidelines is not a score. The term referring to integrated AF management should be called the ABC pathway.

We have modified the text as suggested by the reviewer. Line 54-55.

Best regards

Round 2

Reviewer 1 Report

The paper has been improved with more caution about the conclusions and other edits.  While the outcomes are plausible and statin therapy may well be advisable in this group of patients in any case, the evidence provided by this (retrospective and observational) study is not a conclusive enough basis to make a strong therapeutic recommendation.